# LagLLM: LLM-empowered lead–lag dependency learning for spatial-temporal time series forecasting

**Binqing Wu** [1 2]   **Jian Zhou** [1 2]   **Zongjiang Shang** [1 2]   **Ling Chen** [1 2]

## Abstract

Spatial-temporal time series forecasting is challenging due to complex lead-lag dependencies, which are often ignored or inadequately modeled by existing methods. Thus, we propose LagLLM, the first LLM-empowered framework that explicitly models lead–lag dependencies by unifying data-driven dynamics modeling and knowledge-driven semantic reasoning. Specifically, LagLLM constructs a lead-lag graph by integrating learnable embeddings, spatial proximity, and prompt-guided reasoning from a frozen LLM, which can capture lead-lag dependencies informed by the underlying data structure and semantic knowledge. In addition, LagLLM introduces structural token sorting based on the graph, which can make a fine-tuned LLM explicitly perceive directional and delayed interactions. Experiments on eight real-world datasets show that LagLLM achieves the state-of-the-art performance with improved accuracy, robustness, and interpretability. The code is available at https://github.com/w2obin/LagLLM-2026.

## 1. Introduction

Spatial–temporal time series forecasting is fundamental to many real-world applications (Jin et al., 2024a), e.g., traffic control (Jiang et al., 2023b; Wu et al., 2023a), environment monitoring (Wu et al., 2024; 2026). These data evolve over time while interacting across spatial structures, e.g., road networks or lat–lon grids (Liang et al., 2024). Accurate forecasting is critical for understanding spatial-temporal system behaviors and enabling proactive decision-making. Despite its practical importance, due to inherently complex spatial and temporal dependencies, the task remains highly challenging.

Recent advances attempt to address this challenge using deep neural frameworks, including convolutional networks (Hu et al., 2023; Liu et al., 2022; Wu et al., 2023b), recurrent architectures (Lin et al., 2023; Huang et al., 2024), graph neural networks (GNNs) (Shang et al., 2024; Wu et al., 2025b;a), transformer-based models (Qiu et al., 2024; Liu et al., 2024c; Qiu et al., 2025), and MLP mixers (Wang et al., 2024a; Zhang et al., 2025b). More recently, large language models (LLMs) have shown strong potential for time series forecasting by designing task-specific prompts and converting time series into language-augmented or tokenized sequences (Zhou et al., 2023; Rasul et al., 2023; Liu et al., 2025b). A few LLM-based methods further enhance spatial awareness by incorporating static spatial structures or high-level geographic semantics (Liu et al., 2024a; Li et al., 2024; Huang et al., 2025). However, most existing methods treat spatial and temporal dependencies as separate components, which prevents them from identifying dependencies crossing spatial and temporal dimensions. For example, temporal variations at one location may influence others with different time delays (Zhao & Shen, 2024; Wu et al., 2025b). We refer to the dependencies across space and time as **lead–lag dependencies**. Ignoring such dependencies may lead to an inaccurate estimation of dependency strength and direction, weakening the model's ability to interpret dynamic evolution.

Some studies try to capture lead–lag dependencies, which mainly fall into three categories: (a) Statistical methods analyze such dependencies via Granger causality, cross-correlation, transfer entropy, etc (Geweke, 1984; Zhou & Sornette, 2007). Although interpretable, they rely on linearity or stationarity assumptions, limiting their comprehensiveness. (b) Domain-specific methods design hand-crafted indicators using expert knowledge (Lehalle & Laruelle, 2018; Parsons et al., 2020), e.g., price–volume curves or microstructure signals in quantitative finance. While providing strong priors, they often require heavy manual engineering and fail to generalize beyond specific applications. (c) Recent deep models learn lead–lag dependencies via memory banks (Jiang et al., 2023c;a), attentions (Yang

[1]State Key Laboratory of Blockchain and Data Security, Zhejiang University, Hangzhou, China [2]College of Computer Science and Technology, Zhejiang University, Hangzhou, China. Correspondence to: Ling Chen <lingchen@cs.zju.edu.cn>.

*Proceedings of the 43rd International Conference on Machine Learning*, Seoul, South Korea. PMLR 306, 2026. Copyright 2026 by the author(s).

et al., 2024; Zhao & Shen, 2024), and graphs (Wang et al., 2024b; Wu et al., 2025b). They can capture complex lead-lag dependencies, but their purely data-driven nature makes them sensitive to noise and prone to spurious patterns.

Since LLMs are pre-trained on large and diverse corpora, they possess broad contextual and cross-domain knowledge. Leveraging such knowledge may better distinguish lead–lag dependencies from noise or random fluctuations, improving generalizability and reducing overfitting. Despite this promise, effectively applying LLMs to model lead–lag dependencies remains non-trivial. (1) **Scarce lead–lag priors**: textual descriptions that explicitly characterize lead–lag dependencies are scarce. Spatial-temporal datasets rarely come with textual annotations or semantic labels describing which variate leads or lags another, making it difficult for LLMs to directly infer such dependencies. (2) **Limited topological awareness**: LLMs are pretrained on text and lack an understanding of spatial structure or directional propagation. However, in many real-world scenarios, lead–lag effects are tightly coupled with the underlying topology. For example, in a traffic system, traffic signals propagate along directed road networks (e.g., upstream to downstream road segments) and exhibit delayed temporal responses.

To this end, we propose LagLLM, the first LLM-empowered framework to model lead–lag dependencies for spatial-temporal time series forecasting. LagLLM bridges data-driven dynamics modeling with knowledge-driven semantic reasoning, enabling LLMs to explicitly perceive how information propagates over time following spatial structures. The main contributions are as follows:

- We introduce the Hybrid Lead-Lag Graph Construction Module that parametrizes the lead-lag graph using spatial and temporal embeddings and refines the graph through spatial proximity and a frozen LLM, guided by a prompt that combines predefined templates with learnable features. This module can infer interpretable lead-lag dependencies considering the underlying data structure and the semantic knowledge.

- We introduce the Structural Token Sorting Module that arranges lead tokens before lag tokens based on the lead-lag graph, temporal positions, and spatial centrality. This module can adapt the LLM to explicitly perceive directional and time-delayed interactions coupled with the topology.

- Extensive experimental results on eight real-world datasets demonstrate that LagLLM effectively captures complex lead–lag dependencies and achieves state-of-the-art (SOTA) performance in accuracy, robustness, and interpretability.

## 2. Related Work

### 2.1. Lead-Lag Dependency Modeling for Time Series

Lead–lag dependencies are crucial in many spatial-temporal systems, e.g., biological, energy, and urban systems (Zhu et al., 2024; Mu et al., 2025; Zhang et al., 2025a). Early works mainly rely on statistical analysis (Geweke, 1984; Granger, 1969) and expert knowledge (Lehalle & Laruelle, 2018; Parsons et al., 2020), but these often lack comprehensiveness and generalization, motivating recent shifts toward data-driven modeling. Some methods capture lead–lag dependencies **implicitly**, where methods do not identify lag structures but consider their effects through spatial-temporal modeling (Jiang et al., 2023b; Cao et al., 2025). For instance, GCRN-style models, e.g., AGCRN (Bai et al., 2020) and DGCRN (Li et al., 2023), interleave graph convolutions within recurrent units to propagate spatial information across time. Memory-based models, e.g., PDFormer (Jiang et al., 2023a) and MegaCRN (Jiang et al., 2023c), quantify delayed effects through prototype matching or learnable memory embeddings. Other methods model lead–lag dependencies **explicitly** by identifying lag structures. For example, some methods connect all locations and time steps via attentions and adaptive GNNs, e.g., TraverseNet (Wu et al., 2022) and FCSTGNN (Wang et al., 2024b), to learn global lead–lag dependencies; others construct spliced spatial–temporal graphs, e.g., STSGCN (Song et al., 2020) and DSTCGCN (Wu et al., 2023a), to capture local lead–lag dependencies (Li & Zhu, 2021; Chen et al., 2022; Zheng et al., 2023). More recent methods combine pre-computed cross-correlation coefficients and neural networks, e.g., LIFT (Zhao & Shen, 2024) and MillGNN (Wu et al., 2025b), to flexibly model lead–lag dependencies while retaining statistical interpretability (Long et al., 2024; Yang et al., 2024). Despite these advances, these deep learning works are purely data-driven, making them sensitive to noise and vulnerable to spurious lead-lag patterns.

### 2.2. LLM-Based Models for Time Series Forecasting

Driven by the strong generalization and contextual reasoning abilities of LLMs, recent studies use LLMs for time series forecasting (Liang et al., 2024; 2025). They are mainly in three directions: (1) Direct adaptation, which transforms time series into LLM-compatible inputs through patching, grouping, embedding, and tokenization (Zhou et al., 2023; Bian et al., 2024; Huang et al., 2025) and then feeds them directly into LLMs. For example, STLLM (Liu et al., 2024a) encodes time series through spatial-temporal embedding and fusion, then inputs them into the LLM backbone for forecasting. STD-PLM (Huang et al., 2025) further captures higher-order inter-series correlations and tokenizes them for input to the LLM, enabling effective forecasting and imputation. (2) Prompt engineering, which designs natural-

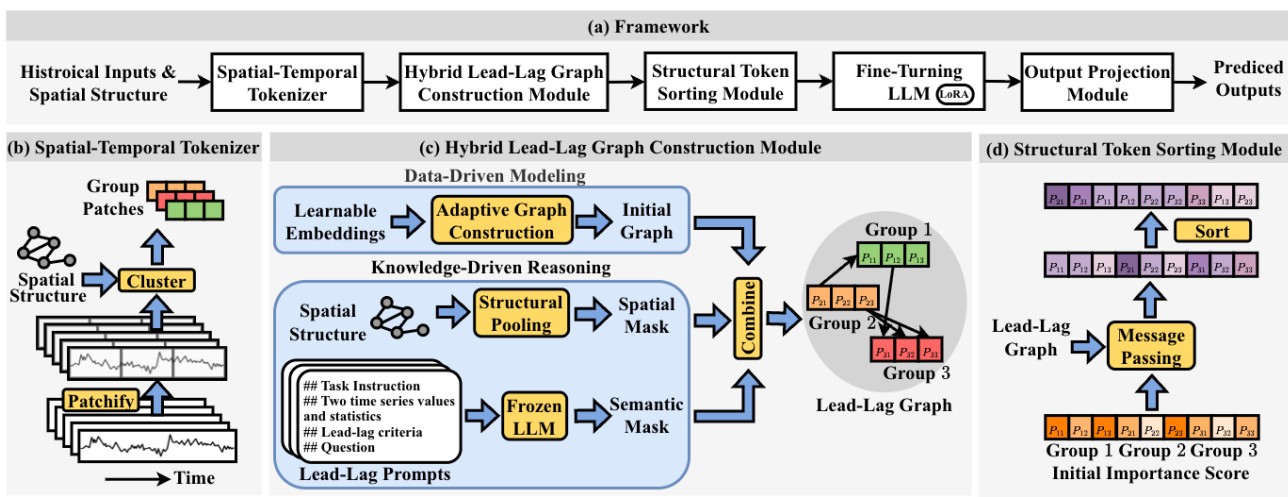

*Figure 1.* Overview of LagLLM.

language or soft prompts to guide frozen or lightly tuned LLMs (Shang et al., 2026; Jiang et al., 2025; Li et al., 2024). For instance, UrbanGPT (Li et al., 2024) uses task descriptions and point-of-interest (POI) information as prompts for spatial–temporal time series forecasting; (3) Cross-modal alignment, which maps time-series embeddings to semantic spaces through attention or contrastive objectives (Liu et al., 2025b; Jin et al., 2024b; Pan et al., 2024). For example, TimeCMA (Liu et al., 2025b) employs cross-attention for dual-modality encoding. These methods experimentally highlight the promise of leveraging LLMs' knowledge and reasoning for time series forecasting. However, they primarily focus on temporal representation alignment and semantic enhancement, while giving limited attention to learning inherent dependencies.

Therefore, we propose LagLLM, the first work to leverage LLMs for learning lead–lag dependencies and improving their ability to perceive how spatial information propagates over time.

## 3. Preliminaries

**Spatial-Temporal Time Series Forecasting.** Spatial-temporal time series are observations indexed by both space and time, represented as $\boldsymbol{X}^{1:T} \in \mathbb{R}^{N \times T}$, where $N$ and $T$ are the number of locations and time steps, respectively. For location $i$, $\boldsymbol{X}_i^{1:T} \in \mathbb{R}^{1 \times T}$ is a time series, and for time step $t$, $\boldsymbol{X}^t \in \mathbb{R}^{N \times 1}$ is a spatial field. In graph-based settings, $\boldsymbol{X}^t$ can be seen as a graph signal on a graph $\mathcal{G}$, with locations as nodes and an adjacency matrix $\boldsymbol{A}_S \in \mathbb{R}^{N \times N}$ encoding spatial dependencies. $\boldsymbol{A}_S$ can be constructed based on criteria, e.g., geographic distance and transportation connectivity.

Given $\boldsymbol{X}^{1:T}$ and adjacency matrix $\boldsymbol{A}_S$, the forecasting task aims to predict future time series $\boldsymbol{X}^{T+1:T+P}$ using a neural

network $\mathcal{F}$ parameterized by $\boldsymbol{\Theta}$. The network is trained to minimize the loss $\mathcal{L}$ between predictions and ground truth, formulated as:

$$\boldsymbol{\Theta}^* = \arg\min_{\boldsymbol{\Theta}} \mathcal{L}(\mathcal{F}(\boldsymbol{X}^{1:T}, \boldsymbol{A}_S; \boldsymbol{\Theta}), \boldsymbol{X}^{T+1:T+P}). \quad (1)$$

**Lead-Lag Dependency.** It refers to the relationship between observations of different time steps and locations. Formally, location $i$ at time step $t_1$ and location $j$ at time step $t_2$ exist a lead-lag dependency if $\boldsymbol{X}_i^{t_1}$ depends on $\boldsymbol{X}_j^{t_2}$, i.e., $\mathbb{P}\left(\boldsymbol{X}_i^{t_1} \mid \boldsymbol{X}_j^{t_2}\right) \neq \mathbb{P}\left(\boldsymbol{X}_i^{t_1}\right)$. The time lag related to this lead-lag dependency is defined as $\tau_{ij} = |t_1 - t_2|$. In this sense, spatial dependencies (i.e., $t_1 = t_2$) and temporal dependencies (i.e., $i = j$) can be viewed as degenerate forms of lead–lag dependencies, where their information propagation occurs only along one dimension.

## 4. Methodology

### 4.1. Overview

The overview of LagLLM is illustrated in Fig. 1. First, the **Spatial-Temporal Tokenizer** segments time series into patches and clusters them based on spatial structure, enriching the tokenization context. Next, the **Hybrid Lead-Lag Graph Construction Module** learns and refines a lead-lag graph using learnable embeddings and a frozen LLM with task-specific prompts, combining data-driven and knowledge-driven elements to capture interpretable dependencies. After that, the **Structural Token Sorting Module** organizes tokens based on the lead-lag graph, temporal position, and spatial centrality, considering the causal nature and spatial topology. These tokens are processed through an LLM backbone with LoRA fine-tuning and passed to the **Output Projection Module** for final forecasts.

## 4.2. Spatial-Temporal Tokenizer

Since lead–lag dependencies arise from interactions spanning time and space, modeling them requires enriching the temporal context of each node and the spatial context provided by its neighbors. Here, each node corresponds to one location associated with a time series.

To tokenize the temporal context, we segment the time series at each node into patches. Each patch serves as a semantic unit that summarizes short-term dynamics and provides a more informative representation for dependency modeling than an individual time step. The representations of each node can be formulated as:

$$\boldsymbol{H}_{\text{Node}} = \text{MLP}(\text{Patchify}(\boldsymbol{X}^{1:T})), \tag{2}$$

where $\boldsymbol{H}_{\text{Node}} \in \mathbb{R}^{N \times N_{\text{p}} \times d}$ is the learned representations. $N, N_{\text{p}}, d$ are the number of nodes, patches, and embedding dimension, respectively.

To tokenize the spatial context, we adopt a learnable assignment matrix $\boldsymbol{S} \in \mathbb{R}^{N_{\text{g}} \times N}$ to cluster nodes. We first randomly initialize the representations of the $N_{\text{g}}$ groups, denoted as $\boldsymbol{H}_{\text{Group}}$. Then, the assignment between groups and nodes is determined via attention scores, which are computed as:

$$\boldsymbol{S} = \text{softmax}(\boldsymbol{Q}\boldsymbol{K}^{\top}/\sqrt{d_{\text{s}}}), \tag{3}$$

where $\boldsymbol{Q}$ and $\boldsymbol{K}$ are the linear projection of $\boldsymbol{H}_{\text{Group}}$ and $\boldsymbol{H}_{\text{Node}}$, respectively. $\boldsymbol{S}$ is optimized during training, assigning each node a soft membership over groups and allowing the network to discover region-level structure without relying on predefined partitions. Since nodes within the same group are expected to share similar regional characteristics, we further introduce a grouping loss to enhance spatial awareness, as defined in Eq. 12.

The series, enriched with spatial and temporal context, is then tokenized as:

$$\boldsymbol{H}_{\text{Token}} = \boldsymbol{S} \cdot \text{Linear}(\boldsymbol{H}_{\text{Node}}), \tag{4}$$

where $\boldsymbol{H}_{\text{Token}} \in \mathbb{R}^{N_{\text{g}} \times N_{\text{p}} \times d}$ represents patch tokens for all groups, where $N_{\text{g}}, N_{\text{p}}, d$ are the number of groups, patches, and embedding dimension, respectively.

## 4.3. Hybrid Lead-Lag Graph Construction Module

To learn the lead-lag dependencies between all patches, we introduce a lead-lag graph, whose structure can be encoded by a weighted adjacency matrix $\boldsymbol{A}_{\text{L}} \in \mathbb{R}^{N_{\text{g}} \times N_{\text{g}} \times N_{\text{p}} \times N_{\text{p}}}$. Each element in $\boldsymbol{A}_{\text{L}}$ quantifies the strength of lead-lag dependencies, where $\boldsymbol{A}_{ij,nm} \neq 0$ represents a lead-lag dependency from the patch $n$ of group $i$ (lead) to the patch $m$ of group $j$ (lag). However, learning such a matrix directly is challenging, as a dense parameterization struggles

| Task Instruction | - **Role**: You are an expert in time-series lead–lag dependency analysis. 
 - **Objective**: Determine whether two time series exhibit a stable and interpretable lead–lag dependency. 
 - **Output Format**: Return "yes" if a stable and interpretable lead–lag dependency exists. Return "no" if the evidence is weak, unstable, contradictory, or ambiguous. Do not include any explanations or extra text. |
|---|---|
| Time Series Statistics | - **First time series**: Observations: [value11, value12, ..., value1n]; Minimum: [min1]; Maximum: [max1]; Mean: [mean1]; Standard Deviation: [std1]; Trend: [trend1] 
 - **Second time series**: Observations: [value21, value22, ..., value2n]; Minimum: [min2]; Maximum: [max2]; Mean: [mean2]; Standard Deviation: [std2]; Trend: [trend2] |
| Lead-Lag Criteria | - Geographical connectivity: [Spatial12] 
 - Lead-lag weights: [LeadLag12] 
 - Consider: cross-correlation, Granger causality, stability, interpretability |
| Question | Based on the provided time-series statistics and lead–lag dependency evidence, determine whether a stable and interpretable lead–lag dependency exists. Answer with "yes" or "no" only. |

*Figure 2.* Illustration of a lead-lag prompt.

to capture the appropriate directionality without inductive bias and faces a risk of over-fitting. Therefore, we learn the lead–lag graph by integrating data-driven modeling with knowledge-driven reasoning.

**Data-Driven Modeling.** Inspired by adaptive graph learning (Bai et al., 2020; Wu et al., 2025b), we use learnable embeddings to parameterize the adjacency matrix, with their inner-product similarity forming a data-driven graph that reflects the underlying structure. Building on this embedding-based formulation, we introduce a spatial embedding $\boldsymbol{E}_{\text{S}} \in \mathbb{R}^{N_{\text{g}} \times d_{\text{e}}}$ to represent the spatial characteristics of each group, and a temporal embedding $\boldsymbol{E}_{\text{T}} \in \mathbb{R}^{N_{\text{p}} \times d_{\text{e}}}$ to represent temporal characteristics at each patch position. We combine these embeddings as:

$$\boldsymbol{E}_{\text{F}} = \boldsymbol{E}_{\text{S}} \oplus \boldsymbol{E}_{\text{T}}, \boldsymbol{A}_{\text{D}} = \text{softmax}(\boldsymbol{E}_{\text{F}}\boldsymbol{E}_{\text{F}}^{T}), \tag{5}$$

where $\oplus$ denotes element-wise addition after broadcasting to matching dimensions. $\boldsymbol{E}_{\text{F}} \in \mathbb{R}^{N_{\text{g}} \times N_{\text{p}} \times d_{\text{e}}}$ encodes inherent characteristics of each patch. $\boldsymbol{A}_{\text{D}} \in \mathbb{R}^{N_{\text{g}} \times N_{\text{g}} \times N_{\text{p}} \times N_{\text{p}}}$ is the initial lead-lag graph.

**Knowledge-Driven Reasoning.** Knowledge-driven lead–lag dependencies are typically sparse and stable in the real world (Löwe et al., 2022; Zhao & Shen, 2024), thus we focus on the top correlated neighbors of each patch and refine the initial lead–lag graph by incorporating explicit structural priors derived from the spatial structure, together with implicit parametric knowledge encoded in an LLM.

*Spatial Mask.* Since lead–lag dependencies strongly depend on the spatial structure, we derive a spatial mask to strengthen lead–lag dependencies among groups that are spatially proximate and strongly connected while mitigating irrelevant ones. The spatial mask is formulated as:

$$\boldsymbol{M}_{\text{S}} = \boldsymbol{S}\boldsymbol{A}_{\text{S}}\boldsymbol{S}^{\top}, \tag{6}$$

where $\boldsymbol{A}_\mathrm{S}$ is the spatial structure between nodes (e.g., distance matrix). $\boldsymbol{M}_\mathrm{S}^g \in \mathbb{R}^{N_\mathrm{g} \times N_\mathrm{g}}$ represents the strength of group connectivity.

*Semantic Mask.* To leverage the knowledge encoded in the parameters of the LLM for lead-lag modeling, we design a lead–lag prompt combining predefined templates with learnable features. Based on prior work on prompt designing for graphs and time series (Zhang et al., 2024; Jin et al., 2024b), the lead–lag prompt consists of four components: (1) Task Instruction, (2) Time Series Statistics, (3) Lead–Lag Criteria, and (4) Question. Notably, the Lead–Lag Criteria mainly include two parts: (i) the learnable part, i.e., the group connectivity from $\boldsymbol{M}_\mathrm{S}$ and lead-lag weights from $\boldsymbol{A}_\mathrm{D}$; and (ii) the predefined part, i.e., cross-correlation, Granger causality, stability, and interpretability, which is not precomputed but simply mentioned to guide the LLM during reasoning. The prompt is passed to a frozen LLM to evaluate whether a lead–lag dependency exists, which allows the LLM to leverage its parametric knowledge effectively. A prompt is shown in the Fig. 2.

To reduce redundant LLM calls, for each group we select the Top-$K$ most correlated neighboring groups based on $\boldsymbol{M}_\mathrm{S}$, such that each group is associated with exactly $K$ neighbors. We then batch the prompts corresponding to each pair of neighboring groups, enabling efficient parallel processing by the LLM. Given the resulting hidden states, the outputs are decoded using a soft classifier head, implemented as a linear layer followed by a softmax function. Finally, we obtain a semantic mask $\boldsymbol{M}_\mathrm{K} \in [0,1]^{N_\mathrm{g} \times N_\mathrm{g}}$, where values closer to 1 indicate a higher likelihood of the presence of a lead–lag dependency. This mask is data-adaptive, dynamically evolving in response to shifting input samples.

The final lead-lag graph, refined by spatial and semantic masks, is formulated as:

$$\boldsymbol{A}_\mathrm{L} = \boldsymbol{A}_\mathrm{D} \otimes (\boldsymbol{M}_\mathrm{S} + \boldsymbol{M}_\mathrm{K}), \tag{7}$$

where $\otimes$ denotes element-wise multiplication after broadcasting to compatible dimensions. $\boldsymbol{A}_\mathrm{L} \in \mathbb{R}^{N_\mathrm{g} \times N_\mathrm{g} \times N_\mathrm{p} \times N_\mathrm{p}}$ captures the directionality and strength of lead–lag dependencies, bridging data-driven and knowledge-driven reasoning.

### 4.4. Structural Token Sorting Module

By sorting patch tokens according to the lead–lag graph, we transform graph-based dependencies into an explicit sequential order that aligns with the sequential modeling paradigm of LLMs. This design enables the LLM to directly capture lead–lag dependencies through token ordering, rather than inferring them implicitly.

Given the lead–lag graph, we evaluate each patch token based on its intrinsic characteristics and its neighbors. Patch

tokens that lead more and occupy more central positions in the graph are assigned higher importance. By placing these more important patch tokens earlier in the input sequence, they can serve as a richer context for all subsequent tokens, thereby maximizing their influence within the LLM's autoregressive attention mechanism.

We initialize the importance of each patch token based on the centrality of groups and the temporal positions of patches. For each group, we first encode each node's spatial centrality based on the eigenvector of the spatial structure $\boldsymbol{V}_\mathrm{N}$, and then cluster them into groups based on the assignment matrix, i.e., $\boldsymbol{B}_\mathrm{G} = \boldsymbol{S} \cdot \mathrm{MLP}(\boldsymbol{V}_\mathrm{N})$, where $\boldsymbol{B}_\mathrm{G} \in \mathbb{R}^{N_\mathrm{g} \times d}$ denotes the spatial centrality of groups. For each patch of group, we embed its temporal positions based on timestamps within the patch $\boldsymbol{T}_\mathrm{p}$, i.e., $\boldsymbol{B}_\mathrm{T} = \mathrm{MLP}(\boldsymbol{T}_\mathrm{p}) \in \mathbb{R}^{N_\mathrm{p} \times d}$. By broadcasting them, we get the initial importance of each patch token, formulated as:

$$\boldsymbol{B}^{(0)} = \mathrm{softmax}(\mathrm{MLP}(\boldsymbol{B}_\mathrm{G} \oplus \boldsymbol{B}_\mathrm{T})), \tag{8}$$

where $\boldsymbol{B}^{(0)} \in \mathbb{R}^{N_\mathrm{g} \times N_\mathrm{p}}$ indicates the importance scores. We use message-passing layers (Rozemberczki et al., 2021) over the lead–lag graph, so that the updated importance of each patch token reflects both its initial score and the weighted influence of its predecessors. The process is formulated as:

$$\boldsymbol{B}^{(l+1)} = \mathrm{MessagePassing}(\boldsymbol{A}_\mathrm{L}, \boldsymbol{B}^{(l)}), \tag{9}$$

where $\boldsymbol{B}^{(l+1)}$ is the updated importance scores. After $L$ layers, sorting tokens by $\boldsymbol{B}^L$ yields the order from most leading to most lagging, formulated as:

$$\hat{\boldsymbol{H}}_\mathrm{Token} = \mathrm{Sorting}(\boldsymbol{H}_\mathrm{Token}, \boldsymbol{B}^L), \tag{10}$$

where $\hat{\boldsymbol{H}}_\mathrm{Token}$ is the sorted tokens.

### 4.5. LLM Fine-Tuning

We use an LLM (i.e., GPT-2) as the backbone. Since layer normalization and positional embeddings in LLMs are highly sensitive to distribution (Zhou et al., 2023; Liu et al., 2025a), we fine-tune these components for adaptation. Due to their small parameter footprints, they are fully updated to align the LLM with the statistics of inputs. In addition, when training resources are sufficient, the attention layers are updated with LoRA (Hu et al., 2022) to model the data structure. Due to the large number of parameters of feed-forward networks, they are frozen.

Since the tokens and their ordering are dynamic, we introduce a learnable set of prefix tokens $\boldsymbol{P} \in \mathbb{R}^{M \times d_\mathrm{p}}$ that act as global information aggregation slots to stabilize representations for forecasting. These prefix tokens are constructed by encoding timestamps and the original time-series values into representations of overall states and trends, following

*Table 1.* Dataset statistics.

| Datasets | #Nodes | Time range | Granularity | Type |
|----------|--------|------------|-------------|------|
| PEMS03 | 358 | 9/1/2018 - 11/30/2018 | 5 min | Traffic Flow |
| PEMS04 | 307 | 1/1/2018 - 2/28/2018 | 5 min | Traffic Flow |
| PEMS07 | 883 | 5/1/2017 - 8/31/2017 | 5 min | Traffic Flow |
| PEMS08 | 170 | 7/1/2016 - 8/31/2016 | 5 min | Traffic Flow |
| METR-LA | 207 | 3/1/2012 - 6/30/2012 | 5 min | Traffic Speed |
| PEMS-BAY | 325 | 1/1/2017 - 5/31/2017 | 5 min | Traffic Speed |
| NYCTaxi | 266 | 4/1/2016 - 6/30/2016 | 30 min | Taxi Demand |
| ChinaAQI | 209 | 1/1/2017 - 4/30/2019 | 1h | Air Quality |

prior work (Huang et al., 2025). The input to our backbone LLM is then constructed by concatenating $\boldsymbol{P}$ with the sorted tokens, i.e., $\text{cat}(\boldsymbol{P}\|\hat{\boldsymbol{H}}_{\text{Token}})$. Passing the input through the fine-tuned LLM yields output representations $\boldsymbol{O}_{\text{g}} \in \mathbb{R}^{N_{\text{g}} \times N_{\text{p}} \times d}$.

### 4.6. Output Projection

Given $\boldsymbol{O}_{\text{g}}$, we map the features of $N_{\text{g}}$ groups into $N$ nodes using the assignment matrix $\boldsymbol{S} \in \mathbb{R}^{N_{\text{g}} \times N}$. For each node, the patch-wise features are flattened into a $(N_{\text{p}} \cdot d)$-dimensional vector and projected to the prediction horizon via a linear layer. The process is formulated as:

$$\boldsymbol{O}_{\text{n}} = \boldsymbol{S}^{\top} \cdot \boldsymbol{O}_{\text{g}}, \quad \hat{\boldsymbol{X}}^{T+1:T+P} = \text{MLP}(\text{Flatten}(\boldsymbol{O}_{\text{n}})), \tag{11}$$

where $\hat{\boldsymbol{X}}^{T+1:T+P} \in \mathbb{R}^{N \times P}$ is the final forecasts.

The training loss combines a grouping loss and a prediction loss. The grouping loss promotes spatial coherence, whereas the prediction loss penalizes discrepancies between predictions and ground-truth targets. The loss is formulated as:

$$\mathcal{L}_{\text{total}} = \alpha \cdot \mathcal{L}_{\text{group}} + \mathcal{L}_{\text{pred}},$$

$$\mathcal{L}_{\text{group}} = \|\boldsymbol{A}_{\text{S}} - \boldsymbol{S}^{\top}\boldsymbol{S}\|_{F}, \quad \mathcal{L}_{\text{pred}} = \sum_{t=T+1}^{T+P} \left\| \boldsymbol{X}^{t} - \hat{\boldsymbol{X}}^{t} \right\|, \tag{12}$$

where $\alpha$ is a coefficient. $\|\cdot\|_{F}$ denotes the Frobenius norm, which encourages the group assignments to align with the spatial structure. $\hat{\boldsymbol{X}}^{t}$ and $\boldsymbol{X}^{t}$ denote the predictions and ground-truth at time step $t$.

## 5. Experiments

### 5.1. Experimental Settings

**Datasets.** We evaluate LagLLM on eight open-source datasets, covering four types of spatial-temporal forecasting tasks: (1) traffic flow forecasting on PEMS03, PEMS04, PEMS07, and PEMS08 (Song et al., 2020), (2) traffic speed forecasting on METR-LA and PEMS-BAY (Yu et al., 2018), (3) taxi demand forecasting on NYCTaxi (Liu et al., 2024a), and (4) air quality forecasting on ChinaAQI (Chen et al.,

2023). The statistics of these datasets are summarized in Table 1. We follow the established preprocessing protocols from the original dataset studies.

**Baselines.** We compare LagLLM with competitive baselines over 2 groups, including (1) **STGNNs/Attentions**: STGCN (Yu et al., 2018), ASTGCN (Guo et al., 2019), GraphWaveNet (Wu et al., 2019), AGCRN(Bai et al., 2020), MTGNN (Wu et al., 2020), STSGCN (Song et al., 2020), STFGNN (Li & Zhu, 2021), TraverseNet (Wu et al., 2022), DGCRN (Li et al., 2023), PDFormer (Jiang et al., 2023a), MegaCRN (Jiang et al., 2023c), LIFT (Zhao & Shen, 2024), VCformer (Yang et al., 2024), STSDN (Cao et al., 2025), and MillGNN (Wu et al., 2025b). (2) **Pre-trained model-based methods**: LagLLaMA (Rasul et al., 2023), STGLLM (Liu et al., 2024b), STLLM (Liu et al., 2024a), TimeCMA (Liu et al., 2025b), CrossST (Liu & Zhang, 2025), STLLM+ (Liu et al., 2025a), and STD-PLM (Huang et al., 2025). TraverseNet, LIFT, VCformer, MillGNN, and LagLLaMA explicitly model lead-lag dependencies. Notably, LagLLaMA, a univariate probabilistic model pre-trained on extensive time-series data with a LLaMA-like architecture, captures only lag dependencies within univariate time series. We compare its results using the most probable prediction as the forecast value.

**Settings.** All experiments are conducted on two NVIDIA A100 80GB GPUs. We use the Adam optimizer with initial learning rates of $\{10^{-2}, 10^{-3}, 5 \times 10^{-4}, 10^{-4}\}$. The batch size is set to 32, and the number of training epochs is set to 200 with an early stopping tolerance of 20. The input and prediction lengths are uniformly set to 12 for PEMS03, PEMS04, PEMS07, PEMS08, METR-LA, PEMS-BAY, and NYCTaxi. For ChinaAQI, the input length is 96 and the prediction length is 24. To tune hyperparameters, we leverage the open-source AutoML toolkit NNI (Microsoft, 2021) with its built-in Bayesian optimization to automatically search for optimal configurations while significantly reducing computational cost.

### 5.2. Overall Performance

Table 2 and Table 3 summarize the forecasting results of LagLLM and baselines. **Bold** and Underline denote the best and second best performance, respectively. The results with ∗ are rerun by us to meet our settings using their official codes. Those without ∗ are cited from the respective original papers or peer works (Huang et al., 2025; Wu et al., 2025b). From the results, we can find that: (1) LagLLM achieves the best performance in 22 out of 27 cases and outperforms the strongest baseline (i.e., STD-PLM) by an average of 1.76% in MAE across eight datasets. These results demonstrate the effectiveness of modeling lead–lag dependencies through a hybrid data-driven and knowledge-driven design. (2) LagLLM outperforms the competitive

*Table 2.* Results on PEMS03/04/07/08.

| Dataset | PEMS03 | | | PEMS04 | | | PEMS07 | | | PEMS08 | | |
|---|---|---|---|---|---|---|---|---|---|---|---|---|
| Metric | MAE | RMSE | MAPE | MAE | RMSE | MAPE | MAE | RMSE | MAPE | MAE | RMSE | MAPE |
| STGCN (2018) | 17.55 | 30.42 | 17.34 | 21.16 | 34.89 | 13.83 | 25.33 | 39.34 | 11.21 | 17.5 | 27.09 | 11.29 |
| ASTGCN (2019) | 17.34 | 29.56 | 17.21 | 22.93 | 35.22 | 16.56 | 24.01 | 37.87 | 10.73 | 18.25 | 28.06 | 11.64 |
| GraphWaveNet (2019) | 19.12 | 32.77 | 18.89 | 24.89 | 39.66 | 17.29 | 26.39 | 41.50 | 11.97 | 18.28 | 30.05 | 12.15 |
| AGCRN (2020) | 15.98 | 28.25 | 15.23 | 19.83 | 32.26 | 12.97 | 22.37 | 36.55 | 9.12 | 15.95 | 25.22 | 10.09 |
| MTGNN (2020) | 15.10 | 25.93 | 15.67 | 19.32 | 31.57 | 13.52 | 22.07 | 35.80 | 9.21 | 15.71 | 24.62 | 10.03 |
| STSGCN (2020) | 17.48 | 29.21 | 16.78 | 21.19 | 33.65 | 13.90 | 24.26 | 39.03 | 10.21 | 17.13 | 26.80 | 10.96 |
| STFGNN (2021) | 16.77 | 28.34 | 16.30 | 20.48 | 32.51 | 16.77 | 23.46 | 36.60 | 9.21 | 16.94 | 26.25 | 10.6 |
| TraverseNet (2022) | 15.44 | 24.75 | 16.40 | 19.86 | 31.54 | 14.38 | 21.99* | 35.83* | 9.45* | 15.68 | 24.62 | 10.87 |
| DGCRN (2023) | 15.21* | 26.86* | 15.88* | 19.25* | 31.17* | 12.66* | 22.21* | 35.85* | 9.15* | 15.98* | 25.67* | 9.97* |
| PDFormer (2023) | 14.74 | 25.59 | 15.35 | 18.31 | 29.97 | 12.10 | 19.83 | 32.87 | 8.53 | 13.58 | 23.51 | 9.05 |
| MegaCRN (2023) | 14.91 | 25.96 | 15.50 | 18.78 | 30.67 | 13.11 | 19.85 | 32.71 | 8.38 | 16.17 | 25.26 | 10.87 |
| LIFT(2024) | 17.37* | 28.63* | 17.09* | 19.41 | 32.53 | 12.35 | 23.63* | 37.16* | 11.34* | 15.22 | 23.86 | 10.18 |
| VCformer(2024) | 17.55* | 30.51* | 18.16* | 19.87 | 31.56 | 12.46 | 23.38* | 37.02* | 11.22* | 15.30 | 23.84 | 10.09 |
| MillGNN (2025) | 16.01* | 26.44* | 15.96* | 18.93 | 30.72 | 12.24 | 21.73* | 35.15* | 9.86* | 14.86 | 23.46 | 9.54 |
| STSDN (2025) | 15.65* | 27.72* | 16.64* | 18.40 | 30.41 | 12.21 | 20.08 | 33.73 | 9.29 | 14.71* | 24.44* | 10.50* |
| LagLLaMA (2023) | 16.38* | 27.64* | 17.21* | 23.40* | 36.09* | 21.54* | 24.47* | 38.96* | 15.15* | 15.91* | 25.00* | 18.34* |
| STGLLM (2024) | 15.26 | **24.11** | 15.73 | 20.00 | 32.11 | 13.69 | 21.98 | 35.02 | 9.72 | 15.53 | 24.74 | 10.15 |
| STLLM (2024) | 17.25 | 27.25 | 22.96 | 19.00 | 30.35 | 13.55 | 21.48 | 34.07 | 10.20 | 14.67 | 23.50 | 10.63 |
| TimeCMA (2025) | 16.74* | 25.37* | 16.88* | 21.79* | 32.68* | 15.01* | 25.51* | 38.60* | 12.91* | 17.82* | 26.90* | 11.19* |
| CrossST (2025) | 15.73* | 24.98* | 19.12* | 20.41* | 32.16* | 14.11* | 22.30* | 34.66* | 13.36* | 16.02* | 24.46* | 13.49* |
| STLLM+ (2025) | 15.91* | 27.33* | 17.20* | 18.58* | 30.46* | 13.40* | 20.92* | 34.82* | 9.28* | 14.68* | 23.87* | 10.47* |
| STD-PLM (2025) | 14.59 | 25.36 | 14.92 | 18.16 | 30.21 | 11.89 | 19.25 | 32.84 | 8.06 | 13.31 | 23.19 | 8.84 |
| LagLLM (Ours) | **14.51** | 24.73 | **14.88** | **18.13** | **29.77** | **11.84** | **19.14** | 32.67 | **8.04** | **13.21** | **23.17** | **8.76** |

*Table 3.* Results on METR-LA, PEMS-BAY, NYCTaxi, and ChinaAQI.

| Dataset | METR-LA | | | PEMS-BAY | | | NYCTaxi | | | | | | ChinaAQI | | |
|---|---|---|---|---|---|---|---|---|---|---|---|---|---|---|---|
| | | | | | | | Pick-up | | | Drop-off | | | | | |
| Metric | MAE | RMSE | MAPE | MAE | RMSE | MAPE | MAE | RMSE | WAPE | MAE | RMSE | WAPE | MAE | RMSE | MAPE |
| PDFormer (2023) | 3.50 | 8.05 | 8.65 | 1.69 | 3.76 | 3.83 | 5.32 | 9.22 | 27.68 | 5.09 | 8.76 | 25.68 | 19.82* | 34.17* | 30.75* |
| MegaCRN (2023) | **2.94** | **6.08** | **8.04** | 1.59 | 3.61 | **3.55** | 5.47* | 9.96* | 25.13* | 5.07* | 9.11* | 21.08* | 19.95* | 34.02* | 31.01* |
| STLLM (2024) | 3.13* | 6.37* | 8.54* | 1.60* | **3.58*** | 3.67* | 5.29 | 9.42 | 20.03 | 5.07 | 9.07 | 19.18 | 19.79* | 32.66* | 31.70* |
| CrossST (2025) | 3.17* | 6.42* | 8.75* | 1.75* | 4.25* | 4.10* | 5.58* | 9.78* | 25.63* | 5.29* | 9.60* | 26.71* | 19.31* | 32.54* | 30.68* |
| STLLM+ (2025) | 3.17* | 6.48* | 8.66* | 1.68* | 3.71* | 3.77* | 5.18 | **8.98** | 19.60 | 4.94 | 8.68 | 18.86 | 20.28* | 33.61* | 31.91* |
| STD-PLM (2025) | 3.11* | 6.34* | 8.59* | 1.65* | 3.64* | 4.59* | 5.24* | 9.21* | 19.69* | 4.93* | 8.61* | 18.66* | 20.36* | 33.37* | 32.52* |
| LagLLM (Ours) | 3.09 | 6.39 | 8.42 | **1.58** | 3.60 | 3.65 | **5.14** | 9.00 | **19.42** | **4.88** | **8.56** | **18.40** | **19.14** | **31.80** | **30.23** |

baseline (i.e., CrossST) over 2.27 % on RMSE in ChinaAQI, where wind-driven pollutant transport induces strong regional lead–lag dependencies. By modeling region-specific and time-varying lead–lag dependencies, LagLLM effectively captures asynchronous dynamics that are overlooked by methods relying on synchronous assumptions, thereby providing a more accurate characterization of spatiotemporal pollutant propagation. (3) In contrast, LagLLM performs relatively weaker on METR-LA, where traffic speed dynamics are primarily dominated by localized patterns (e.g., interactions among geographically adjacent regions), while cross-region lead–lag effects are limited. Consequently, the lead–lag dependencies leveraged by LagLLM become less informative, whereas MegaCRN, with its stronger ability to model fine-grained spatial dependencies is better suited to this dataset.

*Table 4.* Results of few-shot performance.

| Ratio | 5% | | | 10% | | |
|---|---|---|---|---|---|---|
| Metric | MAE | RMSE | MAPE | MAE | RMSE | MAPE |
| MegaCRN | 36.03 | 48.65 | 23.09 | 30.15 | 42.91 | 18.74 |
| STD-PLM | 32.48 | 45.47 | 16.98 | 26.37 | 39.66 | 12.52 |
| LagLLM | **30.07** | **43.29** | **14.06** | **23.82** | **36.56** | **11.09** |
| Impro. | 7.42% | 4.79 % | 17.20% | 9.67% | 7.82% | 11.42% |

### 5.3. Few-Shot Performance

We evaluate LagLLM under few-shot settings following STD-PLM (Huang et al., 2025). Specifically, the models are trained using only 5% and 10% of the PEMS07 training data to simulate data-scarce scenarios. As shown in Table 4, LagLLM consistently outperforms competitive baselines and achieves up to 17.20% improvement. We attribute the

advantage of LagLLM under limited supervision to the integration of LLMs with effective prompts, which introduce informative inductive biases and enable the model to identify meaningful dependencies even when training data is scarce.

## 5.4. Ablation Study

**Hybrid Lead-Lag Graph Construction Module (HLLG).** We design variants to evaluate hybrid lead-lag dependency learning: *(i) w/o llg*, removing the lead-lag graph; *(ii) w/o dm*, removing the data-driven mask; and *(iii) w/o km*, removing the knowledge-driven mask. As shown in Table 5, removing HLLG leads to a performance drop, indicating that explicit lead-lag modeling is crucial. Moreover, the hybrid design outperforms *w/o dm* and *w/o km*, suggesting that data-driven and knowledge-driven cues are complementary.

**Structural Token Sorting Module (STS).** We design variants to evaluate token sorting: *(i) w/o sort*, removing the sorting module; and *(ii) ssort*, sorting tokens by spatial topology. As shown in Table 5, removing sorting degrades performance, highlighting the importance of token ordering. Compared with spatial topology sorting, our method achieves superior results, indicating that sorting tokens according to lead–lag structures may better reflect underlying dynamics.

**Grouping Loss.** We design the variant *w/o $\mathcal{L}_{\text{group}}$*, i.e., removing the grouping loss $\mathcal{L}_{\text{group}}$. The degraded performance in Table 6 verifies its effectiveness. This may be because the grouping loss encourages spatially neighboring nodes to form the same group, which helps the model learn more region-aware and locally correlated representations, thereby enhancing spatial awareness.

**Lead-Lag Prompts.** We design two variants, *w/o criteria*, which removes the lead-lag criteria, and *learn*, which adopts fully learnable prompts. The results in Table 6 demonstrate the effectiveness of the lead-lag prompts. This suggests that incorporating multi-criteria lead-lag justifications (i.e., cross-correlation, Granger causality, and non-linear lead-lag signals) may help uncover more informative lead-lag dependencies.

**LLM Backbone.** We design variants with representative LLM backbones under resource constraints, i.e., *LLaMA-3-8B* and *GPT-2-small*. As shown in Table 6, stronger backbones generally lead to better performance. For a fair comparison, we follow existing works (Huang et al., 2025; Liu et al., 2025a) and choose GPT-2-small as the backbone.

## 5.5. Hyperparameter Study

**Effect of Group Number $N_{\text{g}}$.** $N_{\text{g}}$ controls the granularity of structural aggregation. We choose $N_{\text{g}} \in [8, 16, 32, 64, 128]$, and illustrate the results in Fig.3(a).

*Table 5.* MAE Results of ablation studies on HLLG and STS Modules.

| Variant | HLLG Module | | | STS Module | | LagLLM |
|---|---|---|---|---|---|---|
| | w/o llg | w/o dm | w/o km | w/o sort | ssort | |
| PEMS08 | 14.14 | 13.58 | 13.63 | 13.34 | 13.28 | 13.21 |
| ChinaAQI | 19.74 | 19.52 | 19.41 | 19.47 | 19.36 | 19.14 |

*Table 6.* MAE Results of ablation studies on the loss, prompts, and backbones.

| Variant | Loss | Prompt | | Backbone | |
|---|---|---|---|---|---|
| | w/o $\mathcal{L}_{\text{group}}$ | w/o criteria | learn | LLaMA3 | GPT2 |
| PEMS08 | 13.24 | 13.33 | 13.59 | 13.19 | 13.21 |
| ChinaAQI | 19.18 | 19.26 | 19.20 | 18.97 | 19.14 |

The optimal performance is achieved when $N_{\text{g}}$ equals 32 for PEMS08 and 64 for PEMS04. Smaller values of $N_{\text{g}}$ lead to coarser structural representations with insufficient expressive capacity, whereas larger values result in over-fragmentation and increased optimization difficulty.

**Effect of Correlated Number for Semantic Mask $K$.** $K$ determines the sparsity of the semantic mask. We vary $K$ from 1 to 5 with a step size of 1, and illustrate the results in Fig.3(b). The best performance is obtained when $K$ equals 3 for PEMS08 and 2 for PEMS04. Smaller values of $K$ make the semantic mask overly restrictive and may discard useful contextual information, while larger values of $K$ tend to introduce noisy or weakly relevant correlations that adversely affect model performance.

## 5.6. Efficiency Study

We compare LagLLM with other LLM-based models in terms of training time, trainable parameters, GPU memory usage, and forecasting performance. As shown in Table 7, LagLLM achieves a favorable trade-off between efficiency and performance. Compared with STD-PLM, LagLLM incurs a higher computational cost for each epoch but achieves better accuracy and faster convergence. This improvement stems from the explicit modeling of lead-lag dependencies, which enables richer representations for directional and delayed propagation in the real-world.

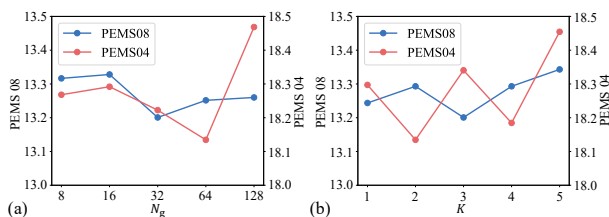

*Figure 3.* MAE Results of hyperparameter studies.

*Table 7.* Results of efficiency study on PEMS04 ($N = 307$).

| Method | Training Time (s/epoch) | # Trainable Parameters (M) | GPU Memory (GB) | MAE |
|---|---|---|---|---|
| STLLM | 11.28 | 42.5 | 5.10 | 19.00 |
| TimeCMA | 75.40 | 29.4 | 21.63 | 21.79 |
| STLLM+ | 42.11 | 3.2 | 12.01 | 18.58 |
| STD-PLM | 16.78 | 3.1 | 11.80 | 18.16 |
| LagLLM | 36.24 | 3.5 | 18.14 | 18.13 |

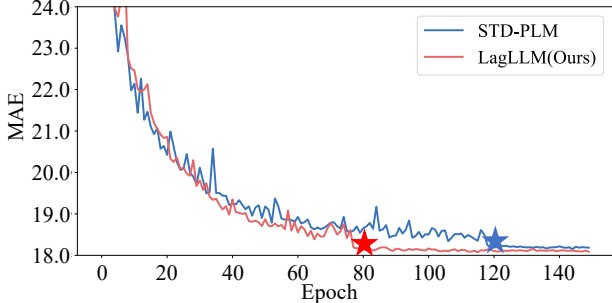

*Figure 4.* Validation loss of STD-PLM and LagLLM on PEMS04. STD-PLM and LagLLM converge at around 120th and 80th epochs, respectively.

**5.7. Case Study**

Fig. 5 presents a case of the learned semantic mask on the ChinaAQI test dataset at 00:00 on December 3, 2018. Fig. 5(a) shows that Group 6 leads Group 13 and Group 11, indicating a directional influence among groups. Fig. 5(b) and Fig. 5 (c) illustrate the grouping results of Group 6, 13, and 11 at 00:00 and 12 hours later, respectively, where circle size denotes AQI magnitude. Pollutants are initially concentrated in Group 6 and subsequently diffuse to Group 13 and Group 11 after 12 hours, which is consistent with the learned semantic mask. This case demonstrates that the semantic mask can capture the directional and delayed propagation of air pollutants in the real world.

## 6. Conclusions and Future Work

In this work, we propose LagLLM, the first LLM-empowered framework for modeling lead–lag dependencies. LagLLM learns a lead–lag graph via a hybrid design

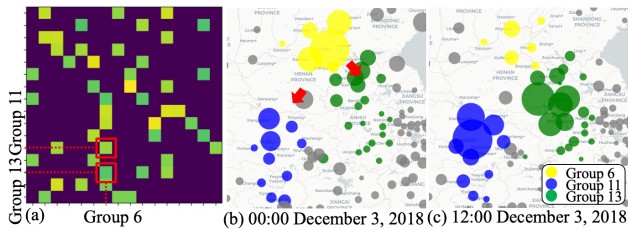

*Figure 5.* Illustration of the learned semantic mask on ChinaAQI.

that combines adaptive graph learning with spatially constrained, prompt-guided refinement from a frozen LLM, and aligns lead–lag dependencies with LLM sequence modeling through structural token sorting. Extensive experimental results demonstrate the SOTA performance of LagLLM on accuracy, robustness, and interpretability. For future work, we will incorporate richer topology and metadata (e.g., road capacity, POIs, and meteorological factors) to enhance modeling capability. Moreover, we will introduce caching for semantic masks and leverage lightweight distilled LLMs to further alleviate the efficiency limitations of LagLLM.

## Acknowledgment

This work was supported by the "Pioneer" R&D Program of Zhejiang (Grant No. 2026C01015). The authors also thank the reviewers for their valuable suggestions.

## Impact Statement

This paper presents LagLLM, a framework that advances the field of Machine Learning by introducing a hybrid approach to modeling lead–lag dependencies via LLM-guided graph refinement. Our work enhances the accuracy and interpretability of complex time series analysis, providing a robust foundation for decision-making in dynamic systems. Our work focuses on scientific advancement using publicly available datasets, and we foresee no negative ethical risks associated with this research.

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
