# OpenReview forum: "LagLLM: LLM-empowered lead–lag dependency learning for spatial-temporal time series forecasting"
_ICML.cc/2026/Conference — ICML 2026 regular_

### Official Review · Reviewer_YVPv · 2026-02-23

**Soundness:** 3
**Presentation:** 3
**Significance:** 3
**Originality:** 2
**Overall Recommendation:** 4
**Confidence:** 4

**Summary:**

LagLLM: LLM-empowered lead-lag dependency learning for spatial-temporal time series forecasting" introduces a framework designed to improve forecasting accuracy by explicitly capturing the asynchronous dependencies where events at one spatial location influence another with a temporal delay. The authors argue that conventional models often treat spatial and temporal dimensions separately, thereby overlooking these complex "lead-lag" interactions that are vital in systems like traffic networks and air quality monitoring.

**Compliance With Llm Reviewing Policy:**

Affirmed.

**Final Justification:**

The author addressed my concerns; therefore, I recommend weak accept.

**Key Questions For Authors:**

1. Have you tested the sensitivity of the Semantic Mask to "semantic noise"? Specifically, if you mask the names of the variables (e.g., changing "Traffic Flow" to "Variable X") or provide counter-intuitive metadata, does the LLM’s decision-making collapse?
2. In a standard GPT-2 backbone, positional embeddings are fixed or pre-trained to represent linear time. When you shuffle these tokens based on a graph-derived importance score ($B^L$), how do you ensure the LLM's internal attention mechanism doesn't misinterpret this as a breakdown in temporal causality? Did you compare this against a model that maintains temporal order but uses "soft prompts" to indicate lead-lag status?
3. How does the framework differentiate between genuine "cross-domain knowledge" and "semantic hallucinations"—where the LLM asserts a lead-lag relationship based on plausible-sounding but factually incorrect reasoning? If the LLM produces a "false positive" (hallucinating a dependency that violates physical laws) or a "false negative" (failing to recognize a dependency due to limited internal data on a specific geography), how does the model recover from this "knowledge poisoning"?

**Limitations:**

yes

**Strengths And Weaknesses:**

Strengths:
1. The author correctly identifies a fundamental limitation in many current spatial-temporal models: the "synchronous assumption". Most existing methods treat spatial and temporal dependencies as separate, parallel components, effectively ignoring how a temporal variation at one node influences another node with a specific time delay.

Weakness:
1. The authors' "Knowledge-Driven Reasoning" module relies on a frozen LLM to validate lead-lag dependencies based on text-based statistics. However, the authors provide the LLM with "predefined templates" and "learnable features." If the LLM is simply rubber-stamping a "yes" or "no" based on statistics that a simple Granger causality test or cross-correlation analysis could determine, the LLM is an expensive, over-engineered heuristic. The authors must prove the LLM adds unique world knowledge that the data itself does not contain.
2. The authors claim that sorting tokens from "most leading to most lagging" allows the LLM to perceive directional interactions. However, LLMs are trained on natural language sequences where order implies logic and grammar, not necessarily causal precedence in a physical system. By rearranging tokens based on a learned graph, the authors may be breaking the temporal continuity the LLM's positional embeddings were designed to handle.
3. To manage complexity, the authors cluster nodes into $N_g$ groups. While computationally efficient, this introduces a massive "aggregation bias." Lead-lag dependencies are often highly localized (e.g., a specific intersection affecting the next). By forcing these into "region-level structures," the authors likely wash out the very fine-grained lead-lag signals the paper claims to capture.

---

> ### Author Rebuttal · Authors · 2026-03-31
>
> We sincerely thank the reviewer for recognizing our key motivation, i.e., identifying the synchronous assumption as a fundamental limitation in many spatio-temporal models. For the technically thoughtful and constructive comments, we carefully address each concern below.
> - Sensitivity of Semantic Mask (Q1)
>   - Following the suggestion, we conduct a variation study about two perturbations: masking variable names leads to minimal performance change (MAE: 18.13 → 18.15 on PEMS08); removing stability and interpretability criteria leads to modest degradation (MAE: 18.13 → 18.21).
>   - These results show that: (1) **Metadata changes affect performance but do not cause collapse**. The model remains robust across both perturbations. (2) **Prompt engineering matters, especially for lead–lag criteria**.
>   - The reviewer's insightful comment has inspired us to further investigate the design of these criteria. We will incorporate this variation study into the revised version and appreciate the reviewer's contribution in the acknowledgement part.
> - Positional Embedding & Temporal Causality (W2, Q2)
>   - We respectfully clarify that temporal information is preserved via timestamp embedding (B_T). Moreover, as stated in Section 4.5, we **fully update** positional embeddings, layer normalization, and attention layers (via LoRA). Such a fine-tuning strategy is **not arbitrary** but follows the mainstream of LLM-based time series methods [1,2], which can bridge the gap between pre-trained language semantics and time series structures.
>   - Following the suggestion, we also compare against a variant maintaining temporal order with soft prompts. The results on ChinaAQI show that the variant achieves 19.42 on MAE, while ours achieves 19.14 on MAE. This confirms that explicit reordering **does not confuse the attention mechanism**; rather, it enables the LLM to better capture directional and delayed dependencies inherent in graph-structured temporal causality of MTS.
>
>     [1]  Zhou, T., et al. One fits all: Power general time series analysis by pretrained lm. NeruIPS, 2023.
>
>     [2] Liu, C., et al. ST-LLM+: Graph enhanced spatio-temporal large language models for traffic prediction. TKDE. 2025.
> - Risk of LLM Hallucinations (Q3)
>   - We acknowledge that the risk of LLM hallucinations is a valid concern for **any** LLM-empowered framework. As Table 5 shows, adopting stronger LLM backbones does improve overall performance.
>   - In LagLLM, to distinguish genuine knowledge from hallucinations, the prompt combines three counterbalancing inputs, including (i) data-derived statistics, (ii) domain-specific spatial structure, and (iii) predefined statistical concepts (e.g., Granger causality and cross-correlation). The LLM's reasoning is therefore **grounded in multiple sources**.
>   - To mitigate potential hallucinations, the semantic mask acts as a **soft prior**. While the prompt instructs the LLM to make explicit judgments, we preserve tolerance for uncertainty: the LLM's last hidden states are decoded via a **soft classifier head** (Line 233–235), producing continuous confidence scores to down-weight uncertain dependencies. In addition, the frozen LLM is applied **only to Top-K candidate neighbors**, focusing reasoning on the most plausible candidates and reducing opportunities for spurious connections.
> - Unique Value of LLMs (W1)
>
>   - We respectfully clarify that the LLM uniquely enables **multi-criteria fusion**, **physical plausibility checking**, and **interpretable reasoning**. (1) The prompt instructs the LLM to simultaneously consider multiple sources, e.g., cross-correlation, Granger causality, non-linear lead-lag signals, and spatial structure, yielding more robust lead–lag dependencies. This is evidenced by LagLLM's consistent outperformance over MillGNN (only cross-correlation priors) across seven datasets. (2) Statistical tests cannot assess physical plausibility, while the LLM leverages pre-trained world knowledge (e.g., pollutants follow wind patterns, traffic propagates downstream) to reject statistically significant but physically impossible dependencies. (3) The semantic mask supports natural-language explanations for lead–lag decisions, which are unattainable by statistical baselines.
> - Aggregation Bias (W2)
>   - We acknowledge this trade-off. We also clarify that our grouping is a **soft assignment**: fine-grained effects propagate through the group graph and back to individual nodes rather than being washed out. The grouping loss (Eq. 12) also encourages spatial coherence without forcing distinct lead–lag behaviors into the same group. Given real-world noise, grouping aggregates consistent patterns across related nodes, acting as regularization against spurious correlations.
>
> Thank you again for your constructive feedback! This has helped us significantly improve the quality of our paper. Should you have any further questions, we look forward to discussing them during the discussion period.

---

> > ### Author Rebuttal · Reviewer_YVPv · 2026-04-01
> >
> > I thank the authors for the rebuttal. The new experiments on the semantic mask and temporal reordering (ChinaAQI) effectively resolve my concerns about robustness and causality. Therefore, I recommend weak acceptance.

---

> > > ### Author Response · Authors · 2026-04-03
> > >
> > > Thank you for your thoughtful review and for taking the time to consider our rebuttal! We really appreciate your recognition of our work and are glad that our responses addressed your concerns. Your suggestions further strengthen the paper.

---

### Official Review · Reviewer_4Pd2 · 2026-03-09

**Soundness:** 3
**Presentation:** 3
**Significance:** 4
**Originality:** 4
**Overall Recommendation:** 5
**Confidence:** 5

**Summary:**

This paper proposes LagLLM, a framework that leverages large language models (LLMs) to explicitly model lead–lag dependencies in spatial-temporal time series forecasting. The key idea is to combine data-driven graph learning with knowledge-driven reasoning from a frozen LLM to construct a lead–lag dependency graph. Based on this graph, the authors introduce a structural token sorting module that reorganizes tokens according to directional lead–lag relationships, enabling the LLM backbone to better capture delayed and directional interactions in spatial-temporal systems. Extensive experiments on eight real-world datasets covering traffic flow, traffic speed, taxi demand, and air quality forecasting demonstrate consistent improvements over strong baselines.

**Compliance With Llm Reviewing Policy:**

Affirmed.

**Final Justification:**

Thank you to the authors for the detailed rebuttal. My concerns have been addressed. The ideas of learning lead–lag graphs in MTS and designing token ordering to align them with LLMs are novel and well-supported by experiments. Therefore, I recommend acceptance.

**Key Questions For Authors:**

- How sensitive the model is to different prompt designs or LLM architectures?
- Does the lead–lag structure evolve over time?
- Why token ordering improves the ability of forecasting?

**Limitations:**

Yes

**Strengths And Weaknesses:**

### Strengths

- The most significant contribution of this work is the hybrid lead–lag graph construction mechanism. This design is novel, which  represents a conceptual advance in how LLMs can contribute to structured dependency discovery.
- The sorting module is an interesting design that bridges graph dependencies with sequential modeling in LLMs.
- The experimental is reletively convincing. Especically, the case study on the ChinaAQI dataset demonstrates that the learned lead–lag graph reflects realistic pollution diffusion patterns, providing an interpretability advantage.

### Weaknesses

- The  framework relies on prompt-guided reasoning from a frozen LLM to construct the semantic mask for lead–lag dependency refinement. However, the paper does not provide sufficient analysis regarding how sensitive the model is to different prompt designs or LLM architectures.
- The paper  remains unclear whether the learned lead–lag structure evolves over time or remains relatively stable across different input windows. Additional analysis on the temporal evolution of the learned graph could provide deeper insight into the model’s behavior and interpretability.
- The paper provides limited explanation of why token ordering improves the ability of transformers to model lead–lag dependencies. A deeper discussion or analysis could strengthen the methodological contribution.

---

> ### Author Rebuttal · Authors · 2026-03-31
>
> We sincerely appreciate the reviewer's positive feedback, especially the recognition of our motivation and technical novelty, and are delighted that the reviewer regards our main contributions as a "conceptual advance". For the technically thoughtful and constructive comments, we carefully address each concern below.
>
> - Variants about LLMs and Prompts (W1, Q1):
>
>   - Our ablation study (Table 5) evaluates different LLM backbones (e.g., LLaMA-3-8B and GPT-2-small). The results show that stronger backbones generally lead to slightly better performance (MAE on PEMS08: 13.19 for LLaMA-3-8B vs. 13.21 for GPT-2-small). For fair comparison and efficiency, we adopt GPT-2-small as the default backbone, following prior works [1,2,3].
>   - We conduct a variation study about two perturbations: masking variable names leads to minimal performance change (MAE: 18.13 → 18.15 on PEMS08); removing stability and interpretability criteria leads to modest degradation (MAE: 18.13 → 18.19). These results show that: (1) Metadata changes affect performance but do not cause collapse. The model remains robust across both perturbations, with MAE degradation within 0.06. (2) Prompt engineering matters, especially for lead–lag criteria.
>
>     [1] Jin, M., et al. Time-LLM: Time series forecasting by reprogramming large language models. ICLR. 2024
>
>     [2] Liu C, et al. TimeCMA: Towards LLM-empowered multivariate time series forecasting via cross-modality alignment. AAAI. 2025.
>
>     [3] Shang, Z., et al. Multi-scale hypergraph meets LLMs: Aligning large language models for time series analysis. ICLR. 2026.
>
> - Explanation about Dynamics of  Lead–Lag Structure (W2, Q2)
>   - The lead–lag structure in LagLLM **does evolve over time**, as it is designed to be **data-adaptive** rather than static. Specifically, the semantic mask M_K is recomputed based on the input sample (Section 4.3), as the prompt incorporates time-varying statistics (e.g., trends, distributions, observations). Moreover, the mask employs a soft classifier head, enabling it to generate varying weights across different samples. This means that even when the same pair of entities appears in M_K, the strength of their lead–lag dependency can differ from one sample to another.
>   - Consequently, according to Eq. (7), since the M_K is dynamic, the overall lead–lag graph A_L can capture **time-varying dependencies**. This property allows the model to handle cross-series dynamics.
> - Explanation about Token Ordering (W3, Q3)
>   - Token ordering prioritizes tokens based on three complementary criteria: the graph establishes lead-lag directions, temporal positions preserve within-series dependencies, and spatial centrality highlights structurally influential groups. Together, they arrange tokens that are **causally upstream, temporally coherent, and spatially informative**, enabling the model to focus on what truly matters for forecasting.
>   - Standard transformers arrange tokens chronologically, causing tokens from different series at the same time step to receive nearly identical positional encodings. This symmetry makes it difficult for attention to distinguish cause from effect. To address this, LagLLM reorders tokens from "leading" to "lagging" within each input window (e.g., all leaders first, followed by laggers, while preserving intra-series temporal order). This simple permutation converts the graph-structured lead–lag problem into a sequential dependency that causal attention naturally solves: earlier positions (leaders) can influence later positions (laggers). In addition, the model fine-tunes positional embeddings to learn functional roles rather than absolute or relative time, further aligning its inductive bias with the directional flow of lead–lag dependencies.
>   - The effectiveness of token ordering is empirically validated by our ablation studies, which show a clear performance drop when token ordering is removed or replaced by spatial topology, confirming its critical role in reflecting the underlying dynamics of multivariate time series.
>
> Thank you again for your thoughtful feedback! This has helped us improve the quality of our paper. Should you have any further questions or concerns, we look forward to discussing them during the discussion period.

---

> > ### Author Rebuttal · Reviewer_4Pd2 · 2026-04-01
> >
> > Thank you to the authors for the detailed rebuttal. My concerns have been addressed. The ideas of learning lead–lag graphs in MTS and designing token ordering to align them with LLMs are novel and well-supported by experiments. Therefore, I recommend acceptance.

---

> > > ### Author Response · Authors · 2026-04-03
> > >
> > > Thank you so much for your careful review and constructive feedback! We sincerely appreciate your insightful comments, which have helped us improve the paper.

---

### Official Review · Reviewer_ixLx · 2026-03-12

**Soundness:** 3
**Presentation:** 4
**Significance:** 3
**Originality:** 4
**Overall Recommendation:** 4
**Confidence:** 3

**Summary:**

The paper introduces LagLLM, an LLM-empowered framework designed for spatial-temporal time series forecasting that specifically targets "lead-lag" dependencies—interactions across space and time where effects at one location propagate to another with a delay. The framework unifies data-driven graph learning with knowledge-driven reasoning by using a frozen LLM to refine a lead-lag graph via prompt-guided semantic masks. Furthermore, it proposes a "Structural Token Sorting" module to order tokens based on lead-lag influence and spatial centrality, allowing the fine-tuned LLM backbone to perceive directional dynamics explicitly through its autoregressive attention mechanism. Experimental results across eight datasets show SOTA performance and improved robustness, particularly in data-scarce (few-shot) scenarios.

**Compliance With Llm Reviewing Policy:**

Affirmed.

**Key Questions For Authors:**

1. Scalability of Semantic Masking: Your current implementation selects Top-K neighboring groups to refine the graph. As the number of nodes (N) or groups ($N_g$) grows significantly in larger city-wide deployments, how do you expect the LLM-calling bottleneck to affect total system throughput?
2. Robustness of Token Sorting: The Structural Token Sorting is dynamic based on input samples. Have you evaluated whether sudden sensor noise or data gaps could cause unstable token ordering, and how that might affect the LLM backbone's output consistency?
3. Inference Latency: While training efficiency is discussed, can you provide metrics on inference latency? For real-time applications like traffic signal control, the end-to-end delay (including LLM-guided graph refinement) is a critical system metric.

**Limitations:**

yes

**Strengths And Weaknesses:**

Strengths:
Soundness: The research is technically grounded in the limitations of existing STGNNs and provides a clear mathematical definition of lead-lag dependencies. The methodology is complete, spanning from patch-based tokenization to a sophisticated hybrid graph construction that effectively balances learnable embeddings with LLM-based priors.

Presentation: The paper is exceptionally well-structured and clear. Figure 1 provides a cohesive architectural overview, and Figure 2 serves as an excellent practical reference for the prompt engineering logic.

Significance: The motivation is highly practical. By modeling asynchronous dynamics—such as pollutant transport or traffic propagation—the system addresses real-world delays that synchronous models often miss.

Originality: The idea of using "token sorting" to translate graph-based lead-lag structures into a sequential order that aligns with an LLM's inherent modeling paradigm is a creative and insightful contribution.

Weaknesses:
System Efficiency and Complexity: From a systems perspective, the reliance on a frozen LLM for semantic mask generation introduces significant computational overhead. While the authors use Top-K filtering and parallel batching to mitigate this, the training time (36.24 s/epoch) and GPU memory usage (18.14 GB) remain substantially higher than competitive baselines like STD-PLM.

Narrow SOTA Gaps in Specific Scenarios: On datasets where localized patterns dominate and cross-region lead-lag effects are limited (e.g., METR-LA), the performance improvement is marginal, suggesting the model's complexity may not always justify the resource expenditure for every application.

Dependency on Pretrained Knowledge: The "Semantic Mask" performance relies heavily on the LLM's ability to reason about specific geographic/domain-specific metadata, which may be a bottleneck if such metadata is unavailable for new, niche datasets.

---

> ### Author Rebuttal · Authors · 2026-03-31
>
> We sincerely appreciate the reviewer's positive feedback, especially the recognition of our work's motivation, technical novelty, and presentation! For the technically thoughtful and constructive comments, we carefully address each concern below.
> - Scalability of Semantic Masking (Q1)
>   - We employ a hierarchical sparse design to ensure **linear scalability**. By grouping, the lead–lag semantic mask operates at the group level rather than the node level. The number of groups N_g is a controllable hyperparameter and is significantly smaller than the number of nodes. Under **the Top-K sparse strategy**, each group only queries K most correlated neighbors (where K \ll N_g),  yielding O(N_g × K)  LLM calls, which is **an approximately linear growth** with respect to N_g.
>   - Frozen LLM calls for semantic masks are **batched and parallelized**. During inference, LLM calls can be optimized such that their overhead is **largely decoupled** from real-time system throughput. LagLLM supports **offloading mask construction** for real-world deployments to minimize runtime overhead, e.g., caching the tokenization of global metadata or triggering updates only by statistical shifts.
> - Robustness of Token Sorting (Q2)
>   - The **structured importance scoring** and **hierarchical grouping** ensure that token ordering remains stable, preserving output consistency. Specifically, token importance is derived from **multiple structure sources** (i.e., noise-insensitive spatial centrality, deterministic temporal position, and knowledge–enhanced lead–lag graph), rather than raw sensor values. In addition, **grouping strategies** aggregate information across nodes, smoothing local perturbations.
>   - The reviewer's insightful comment motivated us to conduct a perturbation experiment on NYCTaxi. We randomly injected Gaussian noise into 10% of inputs and simulated data gaps by masking two consecutive time steps. Token ordering stability is evaluated using **Kendall’s τ** [1], where τ = 1 indicates identical ordering and τ = 0 indicates no correlation. The results show that Kendall’s τ > **0.56** under moderate perturbations, with only 1.9% MAE degradation. Notably, tokens with distinctly high (top 10%) and low importance preserve their relative order, confirming robustness under realistic noise.
>
>     [1] M. G. Kendall, “A new measure of rank correlation,” Biometrika, vol. 30, no. 1/2, pp. 81–93, 1938.
> - Training Cost and Inference Latency  (W1, Q3)
>   - We acknowledge that LagLLM incurs higher per-epoch time and memory than STD-PLM. However, as shown in Fig. 4, LagLLM **converges earlier** (80 vs. 120 epochs), resulting in **comparable total training time** while delivering better accuracy. The additional cost enables **explicit lead-lag modeling** and interpretable semantic masks, which are significant for real-world applications, e.g., pollutant propagation analysis.
>   - As the reviewer points out, inference latency is critical for applications. We compare LagLLM with STD-PLM on inference latency. To ensure a fair comparison, we include LLM calls for semantic masks in the explicit inference path rather than using offloading strategies. On ChinaAQI, STD-PLM achieves 5.34 s/epoch with an MAE of 20.36, while LagLLM (N_g = 8) achieves 4.82 s/epoch with an MAE of 19.77. **LagLLM achieves better predictive accuracy and latency overhead.** Moreover, we acknowledge that the efficiency of LagLLM can be further improved. Promising directions include caching of semantic masks and leveraging lightweight distilled LLMs. We will discuss these optimizations in the future work section in the revised manuscript.
> - Forecasting Performance (W2)
>   - We agree that on datasets with limited lead-lag effects (e.g., METR-LA), LagLLM's improvement over strong baselines is marginal. As discussed in Section 5.2, this is **expected**, as LagLLM does not force spurious structures when such dependencies are weak. LagLLM's value is more evident where lead-lag dependencies are stronger, e.g., **17.20% improvement** on PEMS07 (few-shot) and **2.27%** better than the best baseline on ChinaAQI. We will discuss suitable scenarios more in the revised manuscript.
> - Dependency on Pretrained Knowledge (W3)
>   - We would like to clarify that the semantic mask uses the LLM's parametric knowledge as a **semantic prior**, not merely external metadata. The mask combines data-derived statistics, spatial structure, and learnable lead–lag signals for flexibility. The frozen LLM suppresses spurious correlations by grounding the mask in its pre-trained understanding of lead-lag dependencies. When domain-specific metadata is unavailable, the model can **fall back** to data-derived statistics and learnable signals alone, ensuring broad applicability.
>
> Thank you again for your valuable feedback! Your insights have helped us significantly improve the quality of our paper. Should you have any further questions or concerns, we look forward to discussing them during the discussion period.

---

### Decision · Program_Chairs · 2026-04-30

**Decision:**

Accept (regular)

**Comment:**

This paper proposes a LLM-based spatial–temporal forecasting framework that explicitly models lead–lag dependencies through a lead–lag graph combining data-driven structure and LLM-guided semantic reasoning. The paper is well motivated and with relatively convincing experimental support. Thus, I recommend acceptance.